# Digital health for chronic disease management: An exploratory method to investigating technology adoption potential

**Vasileios Nittas**[1,2], **Chiara Zecca**[3,4], **Christian P. Kamm**[5,6], **Jens Kuhle**[7], **Andrew Chan**[5], **Viktor von Wyl**[1,8]*

**1** Biostatistics & Prevention Institute, Epidemiology, University of Zurich, Zurich, Switzerland, **2** Department of Health Sciences and Technology, Health Ethics and Policy Lab, ETH Zurich, Zurich, Switzerland, **3** Department of Neurology, Neurocenter of Southern Switzerland, EOC, Lugano, Switzerland, **4** Faculty of Biomedical Sciences, Università della Svizzera Italiana (USI), Lugano, Switzerland, **5** Department of Neurology, Inselspital, University Hospital Bern and University of Bern, Bern, Switzerland, **6** Neurology and Neurorehabilitation Center, Luzerner Kantonsspital, Lucerne, Switzerland, **7** Neurologic Clinic and Policlinic, MS Center and Research Center for Clinical Neuroimmunology and Neuroscience Basel (RC2NB), University Hospital Basel, University of Basel, Basel, Switzerland, **8** Institute for Implementation Science in Health Care, University of Zurich, Zurich, Switzerland

* viktor.vonwyl@uzh.ch

**Data Availability Statement:** The underlying data for this analysis are human research participant

## Abstract

### Introduction

The availability of consumer-facing health technologies for chronic disease management is skyrocketing, yet most are limited by low adoption rates. Improving adoption requires a better understanding of a target population's previous exposure to technology. We propose a low-resource approach of capturing and clustering technology exposure, as a mean to better understand patients and target health technologies.

### Methods

Using Multiple Sclerosis (MS) as a case study, we applied exploratory multivariate factorial analyses to survey data from the Swiss MS Registry. We calculated individual-level factor scorings, aiming to investigate possible technology adoption clusters with similar digital behavior patterns. The resulting clusters were transformed using radar and then compared across sociodemographic and health status characteristics.

### Results

Our analysis included data from 990 respondents, resulting in three clusters, which we defined as the (1) average users, (2) health-interested users, and (3) low frequency users. The average user uses consumer-facing technology regularly, mainly for daily, regular activities and less so for health-related purposes. The health-interested user also uses technology regularly, for daily activities as well as health-related purposes. The low-frequency user uses technology infrequently.

data and in combination potentially identifiable. However, data are available upon reasonable request through a standardized procedure. For further information, contact the senior author or ms-register@ebpi.uzh.ch.

**Funding:** The Swiss Multiple Sclerosis Registry is funded by the Swiss Multiple Sclerosis Society. https://www.multiplesklerose.ch/de/. The funders had no role in study design, data collection and analysis, decision to publish, or preparation of the manuscript.

**Competing interests:** The authors have declared that no competing interests exist.

## Conclusions

Only about 10% of our sample has been regularly using (adopting) consumer-facing technology for MS and health-related purposes. That might indicate that many of the current consumer-facing technologies for MS are only attractive to a small proportion of patients. The relatively low-resource exploratory analyses proposed here may allow for a better characterization of prospective user populations and ultimately, future patient-facing technologies that will be targeted to a broader audience.

## Introduction

The number of consumer-facing health technologies, such as smartphone apps and other digital health devices developed to facilitate chronic disease management is skyrocketing, yet, despite some initial successes, most do not live up to their promises [1,2]. Only a small fraction manages to establish a steady and regular user-base [1]. From a technological perspective, functions such as self-tracking, customization, and reward systems are reported to facilitate regular use, while technical misfits (e.g. fast battery drainage), lacking transparency and the misalignment of available functionalities with a patient's needs are only a few examples for rapid abandonment [3,4]. Roger's diffusion of innovation theory suggests that one of three main determinants of how fast innovation is being adopted are associated with the personalities of individuals, termed as potential "adopters" [5,6].

The digitalization of healthcare and many other aspects of social life, as well as the almost universal coverage of internet-accessible smartphones, enable new dimensions of self-care [7]. The digital health market is valued at over 80 billion US Dollars, expected to surpass the 200 billion by 2026 [8]. The much smaller mobile health market is equally booming, estimated at 4 billion US dollars and expected to quadruple over the next decade [9]. Persons with chronic illnesses have round-the-clock access to relevant information, can remotely interact with peers, engage in knowledge exchange, as well as continuously measure and track their health [7,10].

From an individual perspective, past (or current) technology exposure, such as patterns of daily-life and health-related technology use, are proven to predict the adoption, acceptance, and uptake of new technology [11,12]. That often goes hand in hand with low digital literacy, that is, lacking knowledge and skills to use technology and understand electronically provided information [11]. Such patterns may leave certain population segments excluded from the benefits of digital societies, creating equity gaps, known broadly as the digital divide most [13]. Often, those who are excluded are also those that could benefit the most. Previous research has extensively explored which technological, individual, as well as social and environmental factors, might facilitate or hinder successful technology adoption. Despite this, there have been limited efforts to cluster and quantify the number of potential users based on their technology adoption status and willingness [14,15].

### Case study: Multiple sclerosis

Multiple Sclerosis (MS) is a chronic autoimmune disease involving the central nervous system and causing myelin and neuroaxonal damage [16]. MS course and severity vary among patients. At onset, MS follows a relapsing-remitting pattern in most cases, with intermittent recovery from symptoms while approximately 10–15% of patients show continuous symptom worsening from the onset. Initial relapsing-remitting MS cases often transition into a

continuous symptom worsening phase over years. Disease activity and progression are highly individual and unpredictable [16]. For many, MS leads to gradually developing long-term disability and is a disease that strongly impacts daily life, social and mental health [9]. Digital health and the internet become increasingly important in the management of MS, providing a source of important information, as well as support [17]. People with multiple sclerosis have access to over 100 smartphone apps, ranging from support with the initial diagnosis, conducting tests, peer support, knowledge exchange, and access to news and latest evidence [18]. Over 20% of those apps target self-management, such as the management of medication, including reminders, the reporting of side effects, and communication with pharmacies [18]. Wearables may support the surveillance of disease progression, assess sleep quality, fatigue, and movement deficits, while games to promote digital physical activity, also known as exergames, may aid the neurorehabilitation [17]. Considering the possibilities of digital health to positively impact the lives of people with MS, we use MS as a case study.

### Aims

Our work aims to apply a straightforward and low-resource approach to (1) identify and cluster a target population of MS patients according to their technology exposure and use and based on that (2) compare these clusters "beyond technology adoption", using available sociodemographic and health information. Our broader aim is to provide an alternative, practical way to better understand target populations to either assess/predict the adoption of consumer-facing technologies, tailor new technologies or adjust existing ones. We also hope to provide a tool for identifying population clusters that might not have all the required resources and motivation for effectively utilizing technology, which is an essential step towards counteracting remaining digital divides, among MS patients and beyond. Our approach is guided by the Roger's diffusion of innovation theory [2,20]. Along with that logic and without being able to assess personalities, we use a very short survey of four questions to phenotype our target population into various technology adoption categories.

## Materials and methods

Our study is a nested cross-sectional analysis embedded in a nationwide registry on Multiple Sclerosis. We utilized survey-collected information on sociodemographic factors, health characteristics, as well as self-reported utilization of information technology devices and software from the Swiss MS Registry (SMSR), a nationwide, citizen-science registry for people with MS. The SMSR has been extensively described elsewhere [19]. In brief, any person with MS aged 18 years and older who is either residing or receiving care in Switzerland can join the registry. Participation to the registry is offered via an online survey platform or through paper-based questionnaires, in all three national languages (French, German, Italian). Rolling enrollment into the SMSR started in June 2016, and the study is open-ended and currently ongoing. At enrollment, participants are asked to complete a two-part baseline assessment, with the first assessment part covering key sociodemographic data (age, gender, education, work, and disability insurance status, marital and family status) and health status information (MS type, self-reported symptoms at baseline). Participants can then voluntarily opt-in to join the longitudinal study part of the SMSR, for which they complete a second survey assessment part that collects more detailed information on work, education, disease- and treatment history, and health-related quality of life. In the longitudinal study, participants are presented with semi-annual surveys to update the most important personal and disease characteristics, as well as more detailed information on varying topics such as work, mental health, nutrition, living

with MS, or risk factors for MS. The SMSR was approved by the Ethics Committee of the Canton of Zurich (PB-2016-00894). All participants have provided written informed consent.

## Data sources

In October 2020, the SMSR released a survey dedicated to capturing participant experiences at living with MS during the SARS-CoV-2 pandemic, which was made available to all participants who had completed the at least the first part of the baseline assessment, also including those that do not participate in the longitudinal SMSR study. The SARS-CoV-2 questionnaire was designed to gain insights into the daily life challenges of people with MS during the pandemic and was part of the registry. The questionnaire included a set of four questions on the use of digital tools. The questions were created ad hoc and in collaboration with the digital health advisory board of the Swiss MS Society. The first two were adapted from Ziemssen and colleagues' survey on digital health and introduced in the survey with minor modifications [20]. The first elicited the use frequency of multiple internet-connected devices (PC, tablet, smartphone, smartwatch) on a 7-item Likert scale (daily use to never). The second assessed the areas of technology use (e.g., finding information on MS, contacting healthcare providers, interacting with peers) on a 7-item Likert scale (daily use to never). A third question was added eliciting the type of activities performed with the electronic devices during leisure time (i.e., while not being at work), such as information searches, chatting, video calls, or use of apps, again on a 7-item Liker scale (daily use to never). Finally, the fourth question asked whether additional support is required with the use of internet-connected devices, answered with a yes or no. All questions were complemented by free text comment fields to provide additional information. All questions are provided in S1 File. To be included in this study, participants had to have completed the survey and have non-missing socio-demographic data.

## Analysis

To identify possible clusters of technology adoption, we analyzed the four questions on the use of digital tools with multivariate factorial analyses. Initially, we recorded the responses (frequency information) to a dichotomous outcome of (1) at least weekly use or (2) less than weekly use or never. Our analyses were divided into two steps.

Initially, we conducted an exploratory factorial analysis, with parametrization following the recommendations of Howard [21]. We selected the number of factors based on scree plots, which suggested maintaining two factors (both with Eigenvalues of >1). Next, we rotated the resulting factor analysis using the oblique Oblimin algorithm, minimizing complexity and facilitating interpretability. The factor loadings were scrutinized and selected according to the 0.4–0.2 rule [21]. That is, loadings in the principal (higher) factor should achieve a value of at least 0.4 and differ by at least 0.2 units compared with the other factor. Following this approach to factor variable selection, the factor analysis and rotation were repeated with the selected variable set, generating regression-based individual-level score predictions for both factors.

The second step consisted of the calculation of individual-level factor scorings, aiming to investigate possible clusters with similar digital behavior patterns. The segmentation process relied on the k-means clustering algorithm of the predicted factor scores. The number of groups (k) was determined based on the Calinski-Harabasz pseudo-F [22]. As k-means algorithms are known to be sensitive to starting points, these were selected based on preceding Wards-linkage [23]. The resulting clusters were initially analyzed visually using radar plots with each of the variables used in the factor analysis assigned to a radar dimension. We then compared the cluster's sociodemographic and health status characteristics. Median values and interquartile ranges were used for continuous variables and percentages for categorical

variables. Finally, we conducted a sensitivity analysis by grouping the measured usage outcomes by at least daily vs. less than daily (compared with "at least weekly" in the main analysis), which did not materially alter conclusions (not shown). All analyses were performed using Stata 16.

## Results

The larger SMSR survey was completed by 1039 persons, of whom 990 (95.3%) also responded to the four digitalization questions and were included in our factor analysis. Of these, 924 (89%) had fully completed the baseline assessment, allowing a socio-demographic and health status comparison across clusters. In total, about 84% (n = 832) completed the questionnaires online, and 16% (n = 158) on paper. Our study flow is provided in Fig 1.

Our final sample included 254 (25.7%) male and 736 (74.3%) female respondents. In the selected study sample, the median [interquartile range] years of age and years MS diagnosis were 50 [41; 58] and 2010 [2002; 2015], respectively. Overall, 633 (63.9%) reported relapsing-

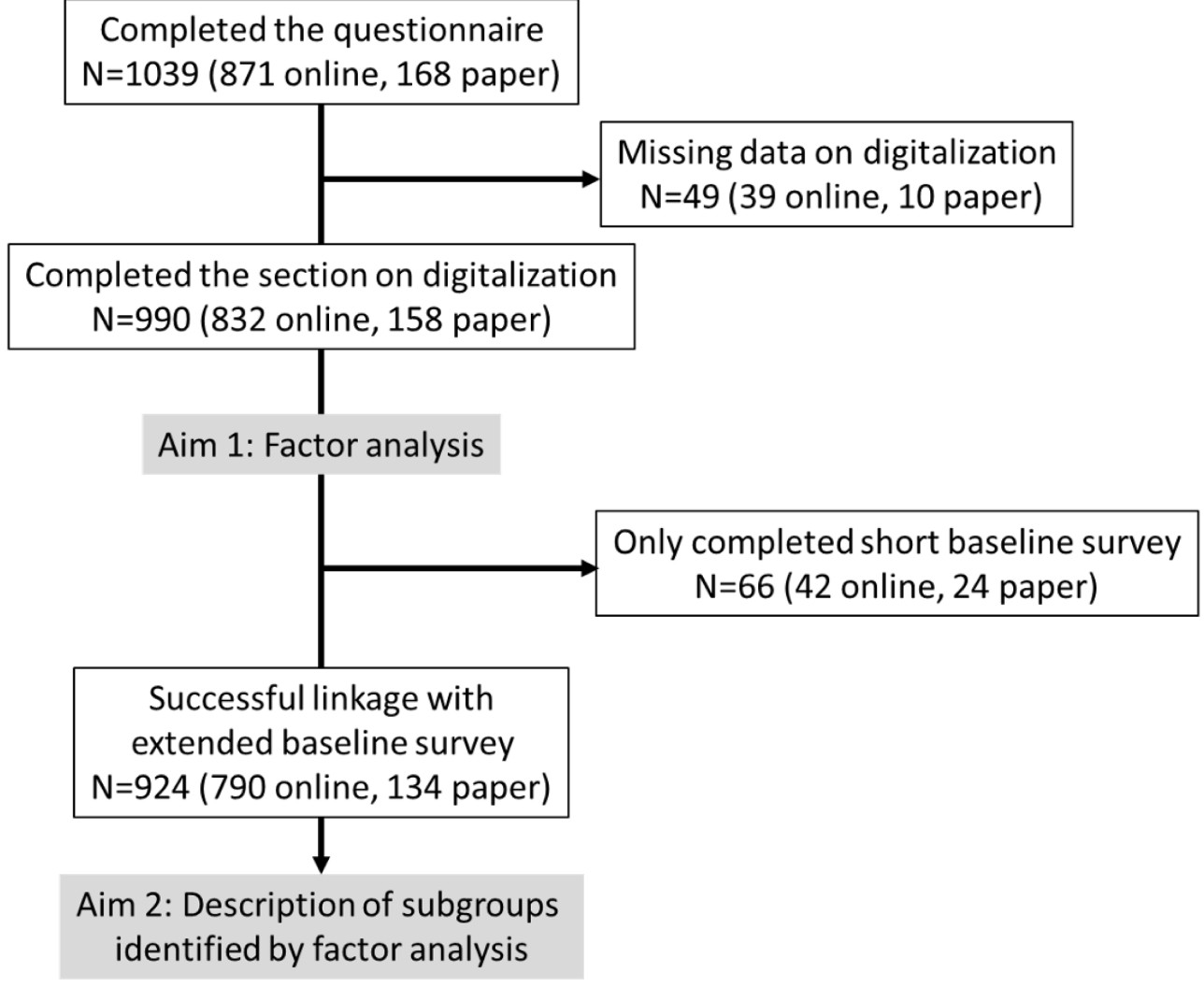

**Fig 1. Study flow chart.**

remitting MS, 108 (10.9%) primary-progressive MS, 171 (17.3%) secondary-progressive MS, and 42 (4.2%) reported transitioning from relapsing-remitting to secondary-progressive MS, and 36 (3.9%) people unclear or missing information on MS type.

## Technology adoption status clusters

Among our 990 respondents, the clustering algorithm suggested three groups of technology adopters, which we defined as (1) average users, (2) health-interested users, and (3) low-frequency users. The clusters are shown in the radar plot of Fig 2, depicting nine activities with sufficient factor loadings and on which the final factor analysis was based (c.f. methods section). These include three variables on technology use (smartphones, mobile and tablet apps, the internet), five variables on the use of internet-connected devices for health-related purposes (searching for healthcare providers, communicating with healthcare providers, searching for MS-relevant information, interacting with peers, making healthcare appointments) and one variable on the use of internet-connected devices for written communication. Table 1 provides an overview of our respondents' sex, age, MS types, and consumer-facing technology use, across the three clusters. The full comparison of the three clusters is provided in S2 File.

## The average user

The largest cluster, defined as the average user group, included 772 individuals (78%). This cluster includes people with MS that regularly (at least weekly) use smartphones, mobile and tablet apps, the internet in general, as well as electronic messaging apps. In contrast, the

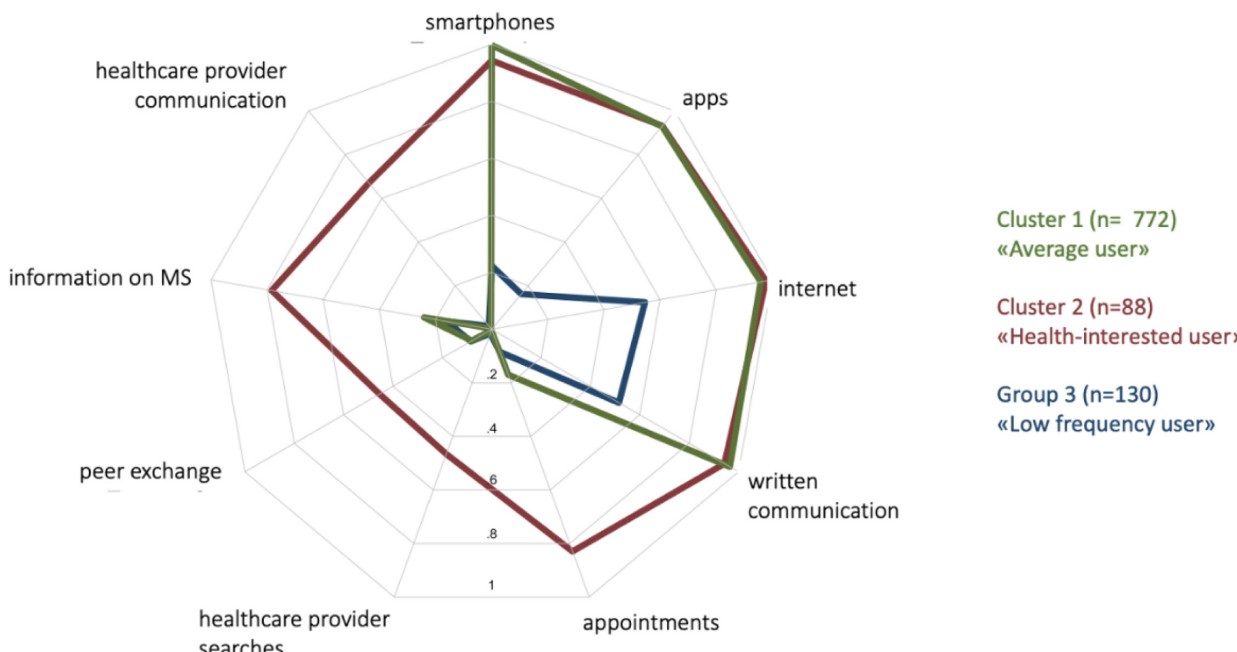

**Fig 2. Radar plot of the three groups derived from the final factor analysis.** Each grey plot line corresponds to a proportion, with the center marking 0% and the outermost line 100%. The colored lines indicate the proportion of respondents in each cluster that report at least weekly use of internet-connected devices and associated health and non-health-related activities.

**Table 1. Cluster comparison.**

| | Cluster 1 average user (N = 772) | Cluster 2 health-interested user (N = 88) | Cluster 3 infrequent user (N = 130) |
|---|---|---|---|
| **Sex (n, %)** **Male** | 195 (25.3) | 17 (19.3) | 42 (32.3) |
| **Female** | 527 (74.7) | 71 (80.7) | 88 (67.7) |
| **Year of birth n, median [interquartile range]** | 761, 1,971 [1,963; 1,980] | 85, 1,974 [1,967; 1,983] | 128, 1,960 [1,952.5; 1,966] |
| **MS type (n, %)** | | | |
| Unknown | 25 (3.2) | 7 (8) | 4 (3.1) |
| Clinically Isolated Syndrome (CIS) | 14 (1.8) | 1 (1.1) | 0 (0.0) |
| Primary progressive | 78 (10.1) | 8 (9.1) | 22 (16.9) |
| Relapsing remitting | 517 (67.0) | 62 (70.5) | 54 (41.5) |
| Secondary progressive | 118 (15.3) | 9 (10.2) | 44 (33.8) |
| Transitional phase | 20 (2.6) | 1 (1.1) | 6 (4.6) |
| **At least weekly use of... (n, %)** | | | |
| PC | 632 (81.9) | 70 (79.5) | 83 (63.8) |
| smartphone* | 770 (99.7) | 83 (94.3) | 29 (22.3) |
| smartwatch | 95 (12.3) | 10 (11.4) | 5 (3.8) |
| tablet | 328 (42.5) | 36 (40.9) | 33 (25.4) |
| apps* | 718 (93.0) | 82 (93.2) | 21 (16.2) |
| internet* | 740 (95.9) | 86 (97.7) | 71 (54.6) |
| **At least weekly electronic device use for... (n, %)** | | | |
| verbal communication | 291 (37.7) | 51 (58.0) | 13 (10.0) |
| written communication* | 745 (96.5) | 83 (94.3) | 67 (51.5) |
| appointments* | 131 (17.0) | 73 (83.0) | 11 (8.5) |
| healthcare provider searches* | 3 (0.4) | 41 (46.6) | 2 (1.5) |
| peer exchange* | 62 (8.0) | 40 (45.5) | 11 (8.5) |
| general information | 445 (57.6) | 76 (86.4) | 41 (31.5) |
| health care provider communication* | 3 (0.4) | 59 (67.0) | 2 (1.5) |
| information on MS* | 186 (24.1) | 69 (78.4) | 21 (16.2) |
| self-tracking | 83 (10.8) | 31 (35.2) | 3 (2.3) |
| **Would like to receive more support (n, %)** | 30 (3.9) | 12 (13.6) | 15 (11.5) |

*indicates grouping variables.

average user group scored low in the use of internet-connected devices for health-related purposes. On a (at least) weekly basis, only less than 30% use such devices to search for MS-related information, about 20% for healthcare appointments, less than 20% to interact with people with MS, and even less for the remaining health-related activities, such as communicating with and searching for healthcare providers. Although not shown in the radar plot as these variables were not identified as cluster-relevant variables, 81.9% of the average users were using Computers/PCs, 42.5% tablets, and 12.3% smartwatches on weekly basis (see Table 1 or S2 File).

## The health-interested user (adopters)

The second cluster, defined as the health-interested user, consists of 88 (8.9%) people with MS. It includes respondents that have a very similar activity profile to those of the average user cluster, however, with more frequent use of the internet and internet-connected devices for health-related activities. On a (at least) weekly basis, about 80% were using such devices to

search for MS-related information, as well as to make appointments with their healthcare providers. Similarly, around 60% used electronic devices for communication with their health care providers and about 50% for interactions with fellow persons with MS, as well as for searching for healthcare providers. Although not shown in the radar plot as these variables were not identified as cluster-relevant variables, 79.5% of the health-interested users were using Computers/PCs, 40.9% tablets, and 11.4% smartwatches on weekly basis (see Table 1 or S2 File].

### The infrequent user

The third identified cluster consists of 130 (13.1%) persons with MS, all of whom report low frequency of internet and internet-connected device use. On a (at least) weekly basis, only about 50% use the internet (for any purposes) or internet-connected devices for written communication (mostly with PC, not shown in plot). Smartphones and apps are used at least weekly by about 20% of the cluster members, while only very few use electronic devices for health-related purposes. Although not shown in the radar plot as these variables were not identified as cluster-relevant variables, 63.8% of the health-interested users were using Computers/PCs, 25.4% tablets, and 3.8% smartwatches on weekly basis (see Table 1 or S2 File).

### Beyond "technology adoption"

Having identified the three adoption status clusters, we considered as equally important to describe those clusters beyond technology adoption, looking at potential socio-demographic and health status differences. This is an essential descriptive step for fully understanding the segments, including their potential health-, literacy, and digital barriers, as well as motivations, of a target population and appraising the potential for health technology to be adopted. We compared 924 of the 990 persons with MS, only those who completed the baseline assessment and had non-missing socio-demographic information. Compared to the full sample, the 66 individuals with MS that were excluded at this stage did not differ in terms of age and year of diagnosis, however, included slightly fewer males (n = 13, 19.1%) and more persons that used paper questionnaires instead of digital (n = 24, 36.4%).

The average user and health-interested user clusters differed across three health-related variables. The health-interested user cluster reported lower median EQ-5D index (83.2) and visual analog scale (70) scores for health-related quality of life, compared to median scores of 90.7 (EQ-5D) and 80 (visual analog scale) of the average user cluster. In addition, the health-interested user cluster reported a median of 6 symptoms, compared to 3 in the average user clusters.

The low-frequency user cluster differed markedly from both other clusters, socio-demographically as well as health-wise. On average, low-frequency users were older (median year of birth 1960) and participated at higher rates via paper questionnaires (43.1%). They reported more severe self-reported disability scores (21.3%), more regular use of wheelchairs (43.1%), canes/ crutches (29.5%), rollators (18.9%), and were more frequently affected with progressive MS subtypes (secondary-progressive (33.8%)), as well as transitional disease phases (4.6%). Finally, low-frequency users reported lower health-related quality of life, with a median 81.5 EQ-5D index score and 70 visual analog scores, which however only differed from the average-user cluster.

## Discussion

The adoption speed and rate of consumer-facing health technology ultimately depends on a target population's adoption willingness, which in turn depends on multiple factors. Two of

these factors are daily-life and health-related technology exposure or use. Our work provides a straightforward and low-resource approach to clustering a target population according to their technology exposure and use, providing a basis to compare clusters across socio-demographic and health-related characteristics. As a use case, our study focused on a broadly recruited population of individuals with MS participating in the Swiss MS Registry, either online or through paper questionnaires. Below we provide a brief account of our findings and discuss their importance in the context of a theoretical example.

## Technology adoption clusters

Our analysis resulted in three technology adoption clusters, which we defined as (a) average users, (2) health-interested users, and (3) low-frequency users. In terms of technology adoption status, we would define the average user cluster as potentially willing to adopt new technology, however, within the boundaries of daily activities. The average health-interested are also potentially willing to adopt new technology, including for health-related purposes. Finally, the average low-frequency user can be defined as a low adopter and potentially hard to reach. Combined they provide a good picture of how well and fast a new digital health device such as a smartphone app for MS symptom management would be adopted targeting participants of the Swiss MS registry. Most of our respondents are likely not interested at all or do not have the mental and/or physical capacities to handle a regular, generic health app. Developing an app without any knowledge about or input from our target population would likely result in lower adoption and use. However, assuming we were to develop such an app based on our findings, what would the implications and lessons learned to be?

## Implications and lessons learned

Although most of our respondents are regularly exposed to at least one internet-connected, consumer-facing technology, primarily smartphones, tablets, PC/notebooks, and less so smartwatches, our target group is not homogeneous at all, especially when it comes to technology use. Developing an app with a homogeneous user group in mind, which comes with certain assumptions, is therefore not the right approach. We have two clusters of people with MS, which also form the majority of our sample, that are likely to be challenging, i.e. those that do not usually use consumer-facing technologies for health-related purposes and those that do not regularly use consumer-facing technologies at all. Only 10% of our sample consists of potentially intrinsic health technology adopters.

Disease burden is likely influencing technology adoption. According to our analysis, the average user cluster has an overall lower disease burden, while the low-frequency user cluster has the highest disease burden. Although counterintuitive, low as well as high disease burden may both inhibit certain types of consumer-facing technology adoption. Low disease burden may keep the motivation of health-related technology use limited, while higher burden might add other hurdles (e.g., physical, mental) that can inhibit the use of regular consumer-facing devices [15,24]. Thus, when designing an app to be used by people with MS different functionalities should be included to provide added values to differently affected and disabled patients. These include for example functionalities that do not focus only on living with disability (or constantly remind of a potentially upcoming disability) but also on living with MS in a broader sense. At the same time, the app should include functionalities and design characteristics to ensure user-friendliness and overcome some limitations of those with a higher MS burden. Here it might be important to consider developing an app that is compatible with tablets or PC's, entails larger icons and verbal cues, reminders as well as information volumes, enabling

easier usability for those with more severe disabilities (e.g., vision deficiencies, sensory and fine motor difficulties, cognitive impairment), without adding to existing fatigue [24].

A further implication of our findings is the understanding that a considerable proportion of our target population might not have the required digital skills to take advantage of digital health solutions. We saw that the low-frequency adoption cluster was on average older, with a lower number of participants having completed higher education. Both variables are well-documented predictors of lower access, use, and acceptance of technology, contributing to existing digital divide gaps [25,26]. Our work allows us to identify and quantify a cluster of a target population that might be potentially most affected by the digital divide. Of note, the proportion size of the low-frequency user group is likely underestimated due to a pre-existing selection bias in those participating in the MS registry. Nevertheless, our approach allows us to be more targeted in exploring the needs of certain subgroups. Does the app require multiple in-app support functions, videos, or other visual cues for those less digitally fluent? Should the app entail family member and peer support functionalities? None of the questions can be fully answered without understanding the needs of prospective users.

## Is MS technology addressing the needs of persons with MS?

Considering these findings, the question arises of whether current MS-related consumer-facing technology is targeting all persons with MS equitably. Other work, such as the systematic app store search by Giunti and colleagues underline that this might not be the case yet [27]. Most current MS-related apps are developed without the involvement of healthcare agencies, often fail to target essential MS symptoms (e.g., fatigue), do not provide well-structured or user-friendly functions, nor do they allow the involvement of family, friends, or healthcare providers [27]. Salimzadeh and colleagues underline that most of the available MS-related applications lack any usability or utility evidence in persons with MS [18]. This underlines that current MS-related technology might not always address the actual needs of a rather diverse population of persons with MS, further underlining the value of our work.

Importantly, the use of technology does not fulfill everyone's needs. Some individuals with MS simply do not wish to use consumer-facing technologies (e.g., because they do not want their disease continuously recalled). However, it is essential health-related technologies are developed in ways that provide all targeted persons with a relatively equal opportunity to access and utilize them. To achieve that is essential to have a broad understanding of how a future userbase is exposed to technology. Using MS as a case study, we have proposed a practical way to do that.

## Limitations

First, while our work was inspired by Roger's diffusion of Innovation theory, our survey was neither derived, nor geared towards it, and thus, should by no means be viewed as an approach to test it. While Roger suggests three main determinants of how fast innovation is being adopted, including the perception of innovation, contextual factors, and individual personalities, our work merely focused on individuals, without assessing their personalities, but rather their current technology exposure and use. Undoubtedly, there are many other individuals as well as contextual factors that drive technology acceptance and use. Yet, we believe that our simple, practical, and low-resource approach is a good example of how a prospective user population can be explored to assess its technology adoption potential. Second, participation in the registry already demands some basic level of digital and health literacy, thus, our sample might not be representative of those persons with MS who have very little literacy. Yet, we aimed to counteract that and increase the proportion of less digitally native people with MS by

providing the option of paper-based participation. Through that, we believe to have captured some of the less digitally literate. Finally, our case study is based on a sample of MS registry participants and cannot be generalized to all people with MS in Switzerland or elsewhere.

## Conclusions

The digital transformation is expected to keep changing chronic disease management and self-care. Yet, the acceptance and adoption of consumer-facing health technology depends on a multitude of factors, including previous experience, exposure, and use, as well as disease burden, cognitive and physical barriers. Based on the data of a short survey about daily life technology use among people with MS, we applied exploratory factor analysis and provided a low-resource, practical approach to exploring some of these factors. Only about 10% of our sample has been regularly using (adopting) consumer-facing technology for health-related purposes. The remainder of our participants were either categorized as regular non-health-related users or overall infrequent users. These findings let us conclude with two key implications. First, most current consumer-facing health technologies for chronic disease management are likely only accessible and attractive to a small proportion of individuals with MS. Second, whenever possible, relatively low-resource exploratory analyses, such as the one proposed here, may allow for a better characterization of prospective user populations and ultimately, future patient-facing technologies that will be targeted to a broader audience.

## Supporting information

**S1 File. Questionnaire.**
(EPS)

**S2 File. Cluster comparison.**
(DOCX)

## Acknowledgments

We are most grateful to all participants of the Swiss MS Registry who dedicated their time and thereby made a crucial contribution to the present research. Further, our gratitude goes to the Swiss MS Society the continuous support. Members of the Swiss MS Registry: Bernd Anderseck, Pasquale Calabrese, Andrew Chan, Claudio Gobbi, Roger Häussler, Christian P. Kamm, Jürg Kesselring (President), Jens Kuhle (Chair of Clinical and Laboratory Research Committee), Roland Kurmann, Christoph Lotter, Kurt Luyckx, Patricia Monin, Stefanie Müller, Krassen Nedeltchev, Caroline Pot, Milo A. Puhan, Irene Rapold, Anke Salmen, Klaas Enno Stephan, Zina-Mary Manjaly, Claude Vaney (Chair of Patient- and Population Research Committee), Viktor von Wyl (Chair of IT and Data Committee), Chiara Zecca. The Swiss MS Registry is supported by the scientific advisory board of the Swiss MS Society.

## Author Contributions

**Conceptualization:** Vasileios Nittas, Viktor von Wyl.

**Data curation:** Viktor von Wyl.

**Formal analysis:** Viktor von Wyl.

**Investigation:** Viktor von Wyl.

**Methodology:** Vasileios Nittas.

**Resources:** Viktor von Wyl.

**Supervision:** Viktor von Wyl.

**Writing – original draft:** Vasileios Nittas.

**Writing – review & editing:** Vasileios Nittas, Chiara Zecca, Christian P. Kamm, Jens Kuhle, Andrew Chan, Viktor von Wyl.

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
