## [Decision Letter · Decision Letter 0]

22 Dec 2022

PONE-D-22-27667Digital health for chronic disease management: An exploratory method to investigating technology adoption potentialPLOS ONE

Dear Dr. Vasulelos Nittas, 

Thank you for submitting your manuscript to PLOS ONE. After careful consideration, we feel that it has merit but does not fully meet PLOS ONE’s publication criteria as it currently stands. Therefore, we invite you to submit a revised version of the manuscript that addresses the points raised during the review process.

Both reviewers have suggested major revisions to the manuscript. In addition to the comments made,

please make all data underlying the findings in your manuscript fully available, in line with the Journal’s requirement. Also, correct your revised manuscript for typographical or grammatical errors.

Please submit your revised manuscript by *Feb 06 2023 11:59PM* If you will need more time than this to complete your revisions, please reply to this message or contact the journal office at plosone@plos.org. Please include the following items when submitting your revised manuscript:A rebuttal letter that responds to each point raised by the academic editor and reviewer(s). You should upload this letter as a separate file labeled 'Response to Reviewers'.A marked-up copy of your manuscript that highlights changes made to the original version. You should upload this as a separate file labeled 'Revised Manuscript with Track Changes'.An unmarked version of your revised paper without tracked changes. You should upload this as a separate file labeled 'Manuscript'.

We look forward to receiving your revised manuscript.

Kind regards,

Chidozie Emmanuel Mbada, PhD.

Academic Editor

PLOS ONE

Journal Requirements:

a) Did participants provide their written or verbal informed consent to participate in this study?

Reviewers' comments:

Reviewer's Responses to Questions

**Comments to the Author**

1. Is the manuscript technically sound, and do the data support the conclusions?

Reviewer #1: Partly

Reviewer #2: Partly

2. Has the statistical analysis been performed appropriately and rigorously? 

Reviewer #1: Yes

Reviewer #2: Yes

3. Have the authors made all data underlying the findings in their manuscript fully available?

Reviewer #1: No

Reviewer #2: Yes

4. Is the manuscript presented in an intelligible fashion and written in standard English?

Reviewer #1: No

Reviewer #2: Yes

5. Review Comments to the Author

Reviewer #1: Dear author(s),

This manuscript is quite novel and interest to readership on digital health and Telehealth for patients with MS.

The strengths of this paper include a large cohort in the case study, results with clear breakdowns of high, low, and average MS users; and interesting figures with results. However, the paper requires key restructuring including introduction, more clarity of figure description, and narrative use of subheaders.

For example, the the introduction of the paper should consider - Rogers's diffusion of innovation as a starting point, then tech use in the digital age, then specific to patients with MS, and then the purpose of this case study.

I hope the author(s) would consider key points to this review and edit appropriately.

Thank you for this submission and opportunity to review this paper.

Reviewer #2: Digital health for chronic disease management: An exploratory method to investigating technology adoption potential

The study examined a low resource approach of capturing and clustering exposure and adoption of technology among patients with MS. The study has the possibility for a better characterization of prospective user populations and ultimately, future patient-facing technologies for broader audience.

However, the following concerns need to be addressed:

1. Numbering of in-text citation not clear, references numbering were not arranged in ascending order of citation . Authors should do necessary corrections following journals instructions

2. In line 136 authors should specify what the information from Swiss MS Registry (SMSR) was utilized for

3. In line 139; what is “the participation” referring to, is it your study or description of SMSR?

4. Under Materials and Methods subheading, what is the research design adopted in this study? Are there any other criteria for inclusion of participants in this study apart from age? Please provide details

5. In lines 141 to 146, authors should indicate specific duration of the longitudinal survey “semi-annual surveys” for how many years?

6. The extensive baseline survey is noted to contain information from short baseline survey. Please give more information on why this is needed in lines 146 to 148

7. Was the informed consent provided by participants written or oral? Line 149

8. Authors should provide a statement to clarify whether the SMSR survey released in October 2020 for capturing participants’ experiences during the SARS-CoV-2 pandemic was part of this study or an existing survey in the registry. Lines 152 to 153

6. PLOS authors have the option to publish the peer review history of their article (what does this mean?). If published, this will include your full peer review and any attached files.

Reviewer #1: No

Reviewer #2: No

---

## [Author Response · Author response to Decision Letter 0]

3 Feb 2023

EDITOR/JOURNAL REQUIREMENTS 

RESPONSE: Done

a) Did participants provide their written or verbal informed consent to participate in this study?

RESPONSE: Completed 

RESPONSE: Completed 

REVIEWER COMMENTS AND RESPONSES

Reviewer #1: 

Comment: Dear author(s),

This manuscript is quite novel and interest to readership on digital health and Telehealth for patients with MS.

The strengths of this paper include a large cohort in the case study, results with clear breakdowns of high, low, and average MS users; and interesting figures with results. 

However, the paper requires key restructuring including introduction, more clarity of figure description, and narrative use of subheaders. For example, the the introduction of the paper should consider - diffusion of innovation as a starting point, then tech use in the digital age, then specific to patients with MS, and then the purpose of this case study 

I hope the author(s) would consider key points to this review and edit appropriately.

Thank you for this submission and opportunity to review this paper.

RESPONSE: Dear reviewer. Many thanks for the kind words and constructive feedback. Please find our responses to your concerns below. 

We have restructured the introduction according to your suggestions. Please find it on p. 3-5. 

We have also adjusted some of the subheadings to improve clarity, such as SMSR to “Data Sources” (p.7); cluster identification and comparison to “Analysis (p.8)

We have also changed some terms to ensure consistency, such as technology use to technology adoption. Please find these throughout the paper. 

Reviewer #2: 

The study examined a low resource approach of capturing and clustering exposure and adoption of technology among patients with MS. The study has the possibility for a better characterization of prospective user populations and ultimately, future patient-facing technologies for broader audience. 

RESPONSE: Dear reviewer, many thanks for your feedback. Please find our point-to-point replies to all your concerns below.

Comment 1: Numbering of in-text citation not clear, references numbering were not arranged in ascending order of citation. Authors should do necessary corrections following journals instructions

RESPONSE: Thanks for pointing that out. We have corrected the numbering of the references. 

Comment 2: In line 136 authors should specify what the information from Swiss MS Registry (SMSR) was utilized for

RESPONSE: Dear reviewer. The use of the SMSR data is outlined throughout the methods section (lines 165-264). We used the data to classify persons with MS into three prototypical “adoption clusters”. These data came from a dedicated online survey that was developed and released by the SMSR, which has been added to the online appendix. The “adoption clusters” were grouped by use of methods from multivariate statistics (polychoric factor analysis). Once the clusters were established, we compared sociodemographic and health-status characteristics that were also collected by the SMSR during baseline assessments. 

We have tried to further clarify these aspects in the section “Materials and Methods”, where we also provide further details on the data origin and variables used for our analysis. Find all changes in lines 165-264.

Comment 3: In line 139; what is “the participation” referring to, is it your study or description of SMSR? 

RESPONSE: Participation here refers to the registry. We have added and clarified that in line 171. 

Comment 4: Under Materials and Methods subheading, what is the research design adopted in this study? Are there any other criteria for inclusion of participants in this study apart from age? Please provide details 

RESPONSE: We have added the study design in lines 165-166. All inclusion criteria (>18, living or receiving care in Switzerland, complete surveys and have non-missing socio-demographic information) are mentioned in lines 169-172 and 223-225. 

Comment 5: In lines 141 to 146, authors should indicate specific duration of the longitudinal survey “semi-annual surveys” for how many years? 

RESPONSE: We have added the duration, which started in June 2016 and is ongoing. Please find it in lines 172-174. 

Comment 6: The extensive baseline survey is noted to contain information from short baseline survey. Please give more information on why this is needed in lines 146 to 148 

RESPONSE: Thanks for pointing that out. For clarity reasons, we have rephrased and refer to this now as a two-part baseline assessment. The rationale for this separation into a short and extensive baseline was to accommodate persons who are not interested in a longitudinal study participation but are willing to complete a one-time survey. This measure helped to strengthen the representativeness of the SMSR and to better examine potential selection and drop-out effects. Please find the changes in lines 174-184. 

Comment 7: Was the informed consent provided by participants written or oral? Line 149 

RESPONSE: Informed consent was written. We added that in lines 184. 

Comment 8: Authors should provide a statement to clarify whether the SMSR survey released in October 2020 for capturing participants’ experiences during the SARS-CoV-2 pandemic was part of this study or an existing survey in the registry. Lines 152 to 153

RESPONSE: The Covid-19 survey was released outside the usual semi-annual schedule due to the urgency for information on the impact of SARS-CoV-2 on the lives of people with MS, however, was part of the registry and was later used for the purposes of this study. This survey differed from regular SMSR semi-annual surveys by not including the standard status update questions on treatment and health in order to keep the survey brief. However, the SARS-CoV-2 was offered to the same SMSR participants and had a very similar completion rate. We clarify that in line 210.

---

## [Decision Letter · Decision Letter 1]

3 Apr 2023

Digital health for chronic disease management: An exploratory method to investigating technology adoption potential

PONE-D-22-27667R1

Dear Dr. Nittas,

We’re pleased to inform you that your manuscript has been judged scientifically suitable for publication and will be formally accepted for publication once it meets all outstanding technical requirements.

Kind regards,

Chidozie Emmanuel Mbada, PhD.

Academic Editor

PLOS ONE

Additional Editor Comments (optional):

Reviewers' comments:

Reviewer's Responses to Questions

**Comments to the Author**

1. If the authors have adequately addressed your comments raised in a previous round of review and you feel that this manuscript is now acceptable for publication, you may indicate that here to bypass the “Comments to the Author” section, enter your conflict of interest statement in the “Confidential to Editor” section, and submit your "Accept" recommendation.

Reviewer #2: All comments have been addressed

2. Is the manuscript technically sound, and do the data support the conclusions?

Reviewer #2: Yes

3. Has the statistical analysis been performed appropriately and rigorously? 

Reviewer #2: Yes

4. Have the authors made all data underlying the findings in their manuscript fully available?

Reviewer #2: Yes

5. Is the manuscript presented in an intelligible fashion and written in standard English?

Reviewer #2: Yes

6. Review Comments to the Author

Reviewer #2: THE AUTHORS HAVE ADDRESSED COMMENTS RAISED SATISFACRORILY. REVISIONS HAVE BEEN MADE TO THE MANUSCRIPT AND INDICATED APPROPRIATELY

7. PLOS authors have the option to publish the peer review history of their article (what does this mean?). If published, this will include your full peer review and any attached files.

Reviewer #2: **Yes: **Marufat Oluyemisi Odetunde Ph.D

---

## [Editor Report · Acceptance letter]

6 Apr 2023

PONE-D-22-27667R1 

Digital health for chronic disease management: An exploratory method to investigating technology adoption potential 

Dear Dr. Nittas:

I'm pleased to inform you that your manuscript has been deemed suitable for publication in PLOS ONE. Congratulations! Your manuscript is now with our production department. 

Kind regards, 

on behalf of

Dr. Chidozie Emmanuel Mbada 

Academic Editor

PLOS ONE